# A New Method for Bare Permafrost Extraction on the Tibetan Plateau by Integrating Machine Learning and Multi-Source Information

Xiaoyang Li [1,2], Yuhe Ji [2,3,*], Guangsheng Zhou [1,2,3], Li Zhou [2,3], Xiaopeng Li [1,2], Xiaohui He [1,2] and Zhihui Tian [1,2]

1 School of Earth Science and Technology, Zhengzhou University, Zhengzhou 450001, China; lixiaoyang7702@gs.zzu.edu.cn (X.L.); zhougs@cma.gov.cn (G.Z.); lixp@gs.zzu.edu.cn (X.L.); hexh@zzu.edu.cn (X.H.); iezhtian@zzu.edu.cn (Z.T.)
2 Joint Laboratory of Eco-Meteorology, Chinese Academy of Meteorological Sciences, Zhengzhou University, Zhengzhou 450001, China; zhouli@cma.gov.cn
3 State Key Laboratory of Severe Weather, Chinese Academy of Meteorological Sciences, Beijing 100081, China
* Correspondence: jiyh@cma.gov.cn

**Abstract:** Bare permafrost refers to permafrost with almost no vegetation on the surface, which is an essential part of the ecosystem of the Tibetan Plateau. An accurate extraction of the boundaries of bare permafrost is vital for studying how it is being impacted by climate change. The accuracy of permafrost and bare land distribution maps is inadequate, and the spatial and temporal resolution is low. This is due to the challenges associated with obtaining significant amounts of data in high-altitude and alpine regions and the limitations of current mapping techniques in effectively integrating multiple factors. This study introduces a novel approach to extracting information about the distribution of bare permafrost. The approach introduced here involves amalgamating a sample extraction method, the fusion of multi-source remote sensing information, and a hierarchical classification strategy. Initially, the available multi-source permafrost data, expert knowledge, and refinement rules for training samples are integrated to produce extensive and consistent permafrost training samples. Using the random forest method, these samples are then utilized to create features and classify permafrost. Subsequently, a methodology utilizing a hierarchical classification approach in conjunction with machine learning techniques is implemented to identify an appropriate threshold for fractional vegetation cover, thereby facilitating the extraction of bare land. The bare permafrost boundary is ultimately derived through layer overlay analysis. The permafrost classification exhibits an overall accuracy of 90.79% and a Kappa coefficient of 0.806. The overall accuracies of the two stratified extractions in bare land were 97.47% and 96.99%, with Kappa coefficients of 0.954 and 0.911. The proposed approach exhibits superiority over the extant bare land and permafrost distribution maps. It is well-suited for retrieving vast bare permafrost regions and is valuable for acquiring bare permafrost distribution data across a vast expanse. It offers technical assistance in acquiring extended-term data on the distribution of exposed permafrost on the Tibetan Plateau. Furthermore, it facilitates the elucidation of the impact of climate change on exposed permafrost.

**Keywords:** Tibetan Plateau; bare permafrost extraction; Google Earth Engine (GEE); machine learning; multi-source information

## 1. Introduction

The Tibetan Plateau, situated in the southwestern region of China, is the highest plateau globally based on its average altitude [1]. In addition, the Tibetan Plateau is an important ecological reserve and research site for the effects of climate change [2]. Moreover, bare land and permafrost are essential components of the ecosystem of the Tibetan Plateau, and their study can provide insight into critical issues such as surface

processes, ecology, and climate change in the region [3–5]. Examining unoccupied terrain and permanently frozen ground can offer a practical understanding of surface phenomena, ecological conditions, alterations in climate, and other significant concerns within the area. Scholars have conducted separate investigations on bare land and permafrost, with comparatively less emphasis on their combined impact. The Holdridge life zone model and its corresponding definitions of bare land and permafrost have been utilized for this analysis. The present study defines bare permafrost as the soil layer at the surface or at a certain depth below the surface, where the temperature of the soil layer is constantly below 0 °C for two or more consecutive years and contains solid water, and where the vegetation cover on the surface consists of less than twelve percent, resulting in a state of bare land in a non-built-up area [6–8]. Bare permafrost is characterized by both barrenness and permafrost, and its formation and evolution are usually closely related to climate, topography, land use, and other factors, indicating the most severe environmental stress [9].

The accurate extraction of permafrost and bare ground boundaries is the basis of bare permafrost extraction. At present, existing permafrost extraction methods are mainly divided into three categories [10]. The first is the empirical model based on long-term observation practice, such as Jaroslav et al. [11], who proposed the TTOP equilibrium model at the circumpolar Arctic scale based on the remotely sensed LST, ERA temporary climate reanalysis, and land cover information and obtained a permafrost zonal dataset at 1 km in the northern hemisphere. Lu et al. [12] simulated the distribution of permafrost on the Tibetan Plateau using an elevation model and an annual mean surface temperature (MAGT) model, which provided valuable insights into the potential impact of climate change on permafrost. The advantages of this method are its low data requirements, ease of use, and computational efficiency, but the disadvantage is that it does not consider the feedback between the systems, and the model state does not evolve forward over time [13]. The second category is a physical model based on the experimental long-term observation of the permafrost state and its change mechanism and the accumulation of knowledge, such as the SIBCASA [14], the advantage of which is that the model state is continuously dynamically evolving and has strong applicability, but the disadvantage is that its structure is more complex and there are higher data requirements [15]. The third category is statistical learning. Shi et al. [16] established a decision tree model based on elevation, MODIS surface temperature, normalized vegetation index, and advanced microwave scanning of radiometer soil moisture to produce a map of the perennial permafrost distribution in the Tibetan Plateau. Niu et al. [17] used a logistic regression model to synthesize remotely sensed land use maps, MODIS surface temperature, and solar radiation to produce a high spatial resolution perennial permafrost distribution map for the Qinghai–Tibet Engineering Corridor. Aalto et al. [18] used statistical modelling methods to predict the relationship between the surface temperature, as well as climate and local environmental factors, and ALT around the North Pole to obtain a 1 km resolution permafrost distribution map. Ran et al. [19] used statistical learning to obtain a 1 km resolution annual Northern Hemisphere mean ground temperature to predict permafrost distribution. Their main advantage is that they provide rigorous mathematical methods to describe sampling and modeling errors, predict permafrost variables, and quantify their uncertainties, while the disadvantage of their methods is that they require a large amount of ground observation data to train the model and validate the source of the simulation results. Traditional supervised/unsupervised classification, neural network classification, linear spectral decomposition, decision tree hierarchical classification, and index extraction are existing techniques for extracting information from bare land [20]. Syakur et al. [21] introduced the EBBI index for identifying bare land and established a threshold for this index to detect small areas of bare land effectively. Tian et al. [22] used TM images with a spatial resolution of 30 m and adopted the QUEST decision tree method to obtain land cover classification data for 2010 in the study area. Qiao et al. [23] employed visible light, thermal infrared, and vegetation indices based on BP neural networks to extract boundaries. Existing bare land extraction relies primarily on the index method, which has high extraction efficiency; it is a simple method

but has poor generalization. The neural network classification method and linear spectral decomposition method have complicated operations and require high data quality, while the decision tree hierarchical classification method classifies data via decision trees, which is simple and efficient but unsuitable for large-scale bare land extraction. Although the studies mentioned above can extract bare land and permafrost to a certain extent, there are still numerous limitations: on the one hand, while existing permafrost distribution maps have been produced and the extent of delineated permafrost is generally large, the spatial and temporal resolution is low and the importance of each indicator for permafrost extraction is not quantified. On the other hand, existing studies do not examine the bare land on the Tibetan Plateau in detail, and there are no effective indicators to suppress noise during the extraction of bare land.

In order to solve the problem of the lack of high-resolution bare permafrost maps of the Tibetan Plateau, this study investigates the remote sensing feature indicators suitable for bare permafrost extraction for the entire Tibetan Plateau. A robust bare permafrost extraction method based on fusing multi-source information with machine learning has been developed by combining it with the Google Earth Engine (GEE) [24] and its provided machine learning classification method [25], and the importance of each metric in the bare permafrost extraction process is given. Meanwhile, uniform and high-confidence permafrost training samples are extracted from the existing multi-source permafrost products to solve the problem of insufficient existing permafrost sample points, and the permafrost extracted through different image synthesis models are compared and analyzed in order to select the best permafrost extraction method. In addition, a 30 m resolution map of the bare permafrost distribution on the Tibetan Plateau is synthesized.

## 2. Study Region and Data

### 2.1. Study Region

The Tibetan Plateau is located in southwestern China, with an average elevation of about 4320 m above sea level and a total area of $3.08 \times 10^6$ km$^2$ [26]. Bare land consists of sandy land, the Gobi desert, saline land, bare rock, gravel land, and other land cover types, which are mainly distributed in the northern, northwestern, and southern parts of the Tibetan Plateau [8]. Moreover, the Tibetan Plateau is a region with one of the largest areas of permafrost in the world, mainly distributed in the cold zones of high mountains and plateaus. The Tibetan Plateau experiences intense radiation, abundant sunshine, low temperatures, temperature decline with height and latitude, significant daily temperature fluctuations, wet and dry spells, prolonged dry and chilly winters, and mild, refreshing, and rainy summers (Figure 1) [27].

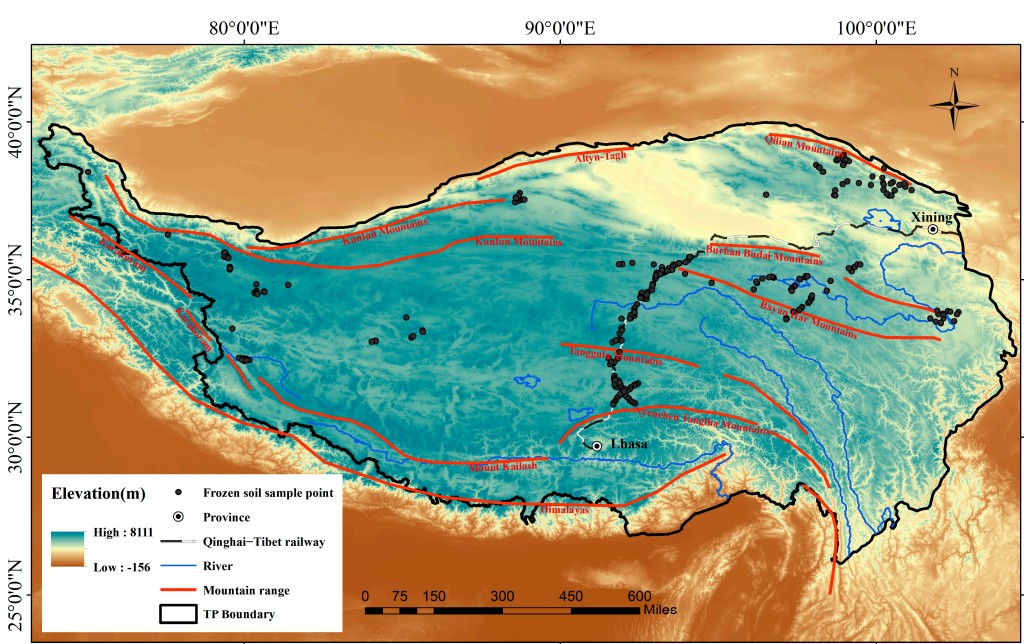

**Figure 1.** Study region (TP Boundary−the boundary of the Tibetan Plateau. Frozen soil sample point−the distribution points of frozen soil on the Tibetan Plateau detected via borehole exploration, pit exploration, soil temperature, and ground−penetrating radar).

### 2.2. Data Sources

#### 2.2.1. Satellite Data

The Landsat data used for the study were released by the United States Geological Survey (USGS), and the images were preprocessed and adjusted. These images contain four visible and near-infrared bands, two short-wave infrared bands, two thermal infrared bands, and two orthorectified surface temperature bands [28].

The MOD10A1.006 Terra Snow Cover Daily Global 500 m dataset provides information on the global condition of snow cover, including the percentage of snow cover, snow depth, and snow water equivalent [29]. It can be utilized to analyze snow season trends and resample them to a resolution of 30 m.

Radar data from the Sentinel-1 satellite synthetic aperture radar (SAR) were provided by the European Space Agency (ESA) and encompass ground-processed and detected C-band SAR data [30]. These data were processed on a logarithmic scale to enhance the dynamic range of the images, as well as the contained multi-polarized and multi-temporal SAR data, and were subjected to thermal noise removal, radiometric calibration, and terrain correction using 30 m elevation data.

#### 2.2.2. Topographic Data

The digital elevation data used in this study were obtained by the Shuttle Radar Topography Mission (SRTM), referred to as SRTMGL1_003. NASA provided the product with a spatial resolution of 30 m [31].

#### 2.2.3. Precipitation Data

The precipitation data are derived from CHIRPS Daily: Climate Hazards Group InfraRed Precipitation With Station Data, which employs infrared satellite data to infer precipitation by analyzing cloud cover temperature and height. In addition, it incorporates actual precipitation data from ground-based weather stations, thereby enhancing the spatial resolution and veracity of the data. It has a daily temporal resolution, a spatial resolution of 0.05 degrees, and a resampling factor of 30 m [32].

### 2.2.4. Permafrost Data

The information on permafrost distribution in the Tibetan Plateau was obtained from borehole surveys, pit probes, soil temperature measurements, and ground-penetrating radar [33–36]. The data were collected from 2015 to 2017 at depths greater than 10 m and were sourced from the existing literature and datasets. Additionally, multi-year permafrost maps were used to obtain permafrost sample points, including maps from top plate temperature models, ground temperature and thermal stability type distribution maps, and 1 km resolution probability maps [37–39].

### 2.2.5. Auxiliary Data

We used two datasets, the Tibetan Plateau land use and land cover datasets, to help extract bare land from areas with permafrost. This allowed us to ensure consistency in our statistical analysis [40–42].

## 3. Extraction Method for Bare Permafrost

This study aims to create a reliable method for extracting bare permafrost using machine learning and information from various sources. We combined existing permafrost data, expert knowledge, and refinement criteria to generate training samples. Then, a hierarchical classification strategy combined with random forest classification was applied to select a suitable vegetation cover threshold to extract the extent of bare ground. Finally, we used layer overlay analysis to determine the bare permafrost boundary. The main steps involved generating training and test samples, constructing features, selecting features, conducting supervised classification, and evaluating accuracy (Figure 2).

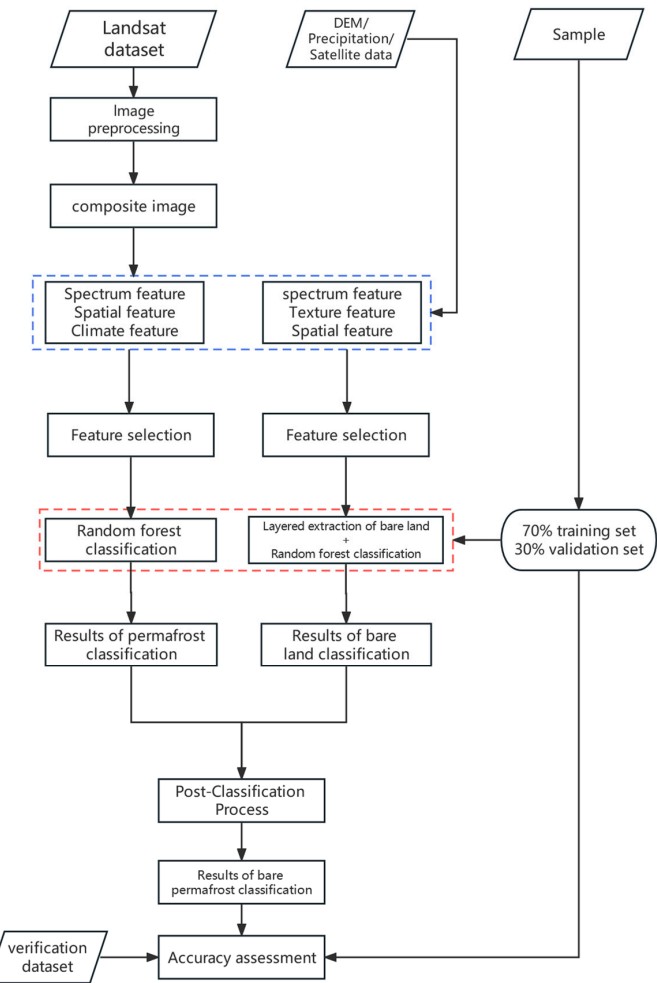

**Figure 2.** The process of bare permafrost extraction.

### 3.1. Image Preprocessing

Image preprocessing aims to eliminate the effects of clouds and seasonal snowfall and synthesize images with fewer clouds and less snow in the study area of the Tibetan Plateau. The main steps in this process include dataset filtering, cloud filtering, and image synthesis. Dataset filtering mainly establishes the research dataset based on the geographic scope and time of the study, and the image dataset from 2015 to 2017 was predominantly used in this study. Cloud filtering was performed using the cloud scoring algorithm provided by the GEE platform, with scores ranging from 0 (no cloud) to 100 (very thick cloud) [43]. After repeated experiments, this study's optimal cloud removal filter threshold was 20%. The median value of the pixel set was used to synthesize the image [44], i.e., the median of the cloud-filtered overlapping pixel values of each pixel was used as the new pixel value to synthesize the complete image of the study area.

### 3.2. Sample Point Generation

This study identified bare land and permafrost separately and then synthesized bare permafrost via overlay analysis. Bare land was classified in a total of four categories as bare land and vegetation, built-up areas, water bodies, and snow and ice, and the sample points were obtained through existing land use information and visual interpretation in order to make the sample points random and uniform [45]. Existing permafrost products with high temporal consistency were used, and these temporally stable permafrost pixels were selected as the primary candidate points. Considering that there is a time interval between previous permafrost products and our study and that permafrost usually follows the pattern of edge-to-center contraction, a morphological erosion and expansion filter with a local window of 3 × 3 was applied to the permafrost products for a morphological "open" operation to further ensure the confidence level of the permafrost training sample [46,47]. Thus, the most significant and most stable boundary of the permafrost was obtained [48] and permafrost sample points were selected (Figure 3). The sample point selection in this study is shown in Table 1.

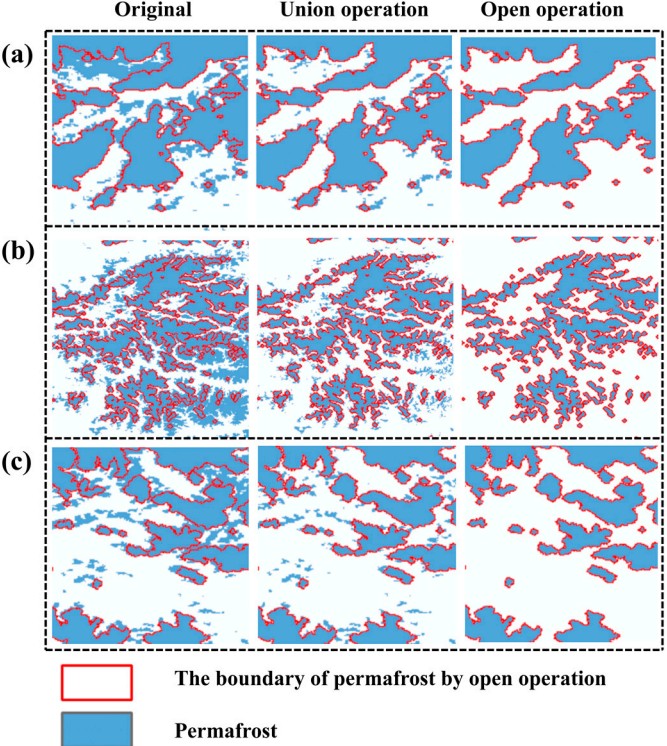

**Figure 3.** (**a–c**) show the distribution of permafrost at different locations on the Tibetan Plateau; from left to right, these are original permafrost distribution, joint operation permafrost distribution, and open operation permafrost distribution, respectively.

**Table 1.** Type and number of sample points.

| Permafrost | | Bare Land | |
|---|---|---|---|
| Permafrost (measured) | 244 | Bare land and vegetation | 845 |
| Permafrost (generation) | 56 | Built-up area | 257 |
| Non-permafrost (measured) | 116 | Water bodies | 304 |
| Non-permafrost (generation) | 113 | Ice and Snow | 373 |

### 3.3. Feature Construction

3.3.1. Spectrum Characteristics

Multiple indices were derived from various combinations of these bands to reflect the characteristics of water bodies, snow and ice, vegetation, and bare land. The employed indices are shown in Table 2.

**Table 2.** Characteristic indices.

| Category | Index Name | Index Characterization |
|---|---|---|
| Vegetation Index | NDVI [49] | $\frac{\rho_{NIR}-\rho_{Red}}{\rho_{NIR}+\rho_{Red}}$ |
| | DVI [50] | $\rho_{NIR}-\rho_{Red}$ |
| | RVI [51] | $\frac{\rho_{NIR}}{\rho_{Red}}$ |
| | EVI [52] | $\frac{\rho_{NIR}-\rho_{Red}}{\rho_{NIR}+6\times\rho_{Red}-7.5\times\rho_{Blue}+1}$ |
| Water Body Index | NDWI [53] | $\frac{\rho_{Green}-\rho_{NIR}}{\rho_{Green}+\rho_{NIR}}$ |
| | MNDWI [54] | $\frac{\rho_{Green}-\rho_{SWIR1}}{\rho_{Green}+\rho_{SWIR1}}$ |
| Snow Index | SWI [55] | $\frac{\rho_{Green}(\rho_{NIR}-\rho_{SWIR})}{(\rho_{Green}+\rho_{NIR})(\rho_{SWIR}+\rho_{NIR})}$ |
| | NDSI$_{Snow}$ [56] | $\frac{\rho_{Green}-\rho_{SWIR2}}{\rho_{Green}+\rho_{SWIR2}}$ |
| Bare land/building index | NDbaI [57] | $\frac{\rho_{Red}-\rho_{TIR1}}{\rho_{Red}+\rho_{TIR1}}$ |
| | SABI [57] | $\frac{\rho_{SWIR}-\rho_{Green}-\rho_{Blue}}{\rho_{SWIR}+\rho_{Green}+\rho_{Blue}}$ |
| | NDSI$_{Soil}$ [58] | $\frac{\rho_{Green}-\rho_{SWIR1}}{\rho_{Green}+\rho_{SWIR1}}$ |
| | NDISI [59] | $\frac{\rho_{TIR1}-(\rho_{Green}+\rho_{NIR}+\rho_{SWIR1})/3}{\rho_{TIR1}+(\rho_{Green}+\rho_{NIR}+\rho_{SWIR1})/3}$ |
| | BI [60] | $\frac{\rho_{SWIR1}+\rho_{Red}-\rho_{NIR}-\rho_{Blue}}{\rho_{SWIR1}+\rho_{Red}+\rho_{NIR}+\rho_{Blue}}$ |

where $\rho_{Green}$, $\rho_{SWIR1}$, $\rho_{SWIR2}$, $\rho_{BLUE}$, $\rho_{NIR}$, $\rho_{RED}$, and $\rho_{TIR1}$ are the reflectance values of Landsat images in the green band (Green), shortwave infrared bands (SWIR1 and SWIR2), blue band (Blue), near infrared band (NIR), red band (Red), and thermal infrared band (TIR1), respectively.

The following formula [61] describes the fractional vegetation cover to indicate vegetation growth in each area:

$$FVC = \frac{NDVI - NDVI_{Soil}}{NDVI_{Veg} - NDVI_{Soil}}, \tag{1}$$

where $NDVI_{Soil}$ is the NDVI value of the pure soil image element; $NDVI_{Veg}$ and Total are the NDVI values of the pure vegetation image element, obtained by intercepting the upper and lower thresholds of the NDVI of the image using 5% confidence level; and NDVI refers to the true NDVI value of the synthetic image pixels.

As K-T transforms classification features, three indices are obtained: the brightness, greenness, and humidity indexes. The brightness index represents the overall reflection effect of the feature, the greenness index represents the surface vegetation condition, and the humidity index represents the surface moisture condition [62].

This study utilizes data from the Sentinel-1 satellite SAR, whose vertical–vertical (VV) polarization band can detect surface texture, morphology, and surface features. The vertical–horizontal (VH) polarization band can detect dispersion signals from objects such as vegetation, snow and ice, surface moisture, sediment, and other information and is sensitive to information such as soil moisture and vegetation structure.

### 3.3.2. Texture Features

Typically, features have distinct texture characteristics; for instance, various types of ground cover typically exhibit distinct texture patterns in remote sensing images. By extracting the image's texture features, the "same feature, different spectrum, different feature, same spectrum" problem can be resolved [63]. The gray-level co-occurrence matrix (GLCM) describes an image's texture by analyzing the relative positional relationships between different gray levels in the image and by calculating the co-occurrence matrix in multiple directions to capture the different texture information. This study includes six standard texture features of synthetic image band ratios: angular second-moment, contrast, correlation, variance, inverse moments, and entropy [64].

### 3.3.3. Topographic Characteristics

Topography is one of the critical soil-forming factors and one of the factors that must be considered in the classification process. Of particular significance is the elevation factor, which directly determines the macroscopic pattern of regional permafrost distribution. The topography and soil type have a close relationship. Desertification, salinization, and marshy soil types are distinguished primarily by the effect of altitude and slope on the water table. In this investigation, elevation, slope, and direction were constructed as three independent bands using DEM data [65] to construct the original features. Simultaneously, latitude and longitude can cause latitudinal zonation caused by the north–south differences in heat and the differences in moisture status, which are a result of varying distances to the ocean, influencing the macroscopic distribution pattern of permafrost [37].

### 3.3.4. Climate Characteristics

This study utilizes mean annual temperature, annual precipitation, and annual snow-pack data as climatic characteristics. Snow cover is a significant factor in reducing the magnitude of ground temperature variability and increasing the mean winter surface temperature because it prevents heat exchange between the surface and the atmosphere and prevents some heat from reaching the surface. Precipitation impacts permafrost's hydrothermal stability by altering the soil's surface energy balance and thermal parameters, thereby altering the freeze–thaw and hydrothermal transport processes [66]. The surface temperature reflects the regional surface radiative heat balance and the characteristics of atmospheric circulation, which influence permafrost changes by acting on the hydrothermal transport processes between the atmosphere and the surface, and the formation and development of permafrost are highly dependent on surface temperature [67]. All features constructed in this study are given in Table 3.

**Table 3.** Characteristic indicators of bare permafrost extraction on the Tibetan Plateau.

| | Category | Features |
|---|---|---|
| Permafrost indicators | Spectrum characteristics | NDVI, EVI, RVI, DVI, SWI, LSWI, NDWI, $NDSI_{Snow}$, greenness, brightness, humidity, FVC, Sentinel-1 VV and VH |
| | Spatial characteristics | Elevation, slope, slope direction, longitude, latitude |
| | Climate characteristics | Annual precipitation, average annual surface temperature, snow cover |
| Bare land index | Spectrum characteristics | B2~B7, NDSI, BI, $NDSI_{Soil}$, SABI, NDVI, NDWI, MNDWI, SWI, $NDSI_{Snow}$, NDbaI, greenness, brightness, humidity |
| | Texture characteristics | Second-order moments, contrast, correlation, variance, inverse moments, entropy |
| | Spatial characteristics | Elevation, slope, slope direction |

### 3.4. Feature Selection

Feature selection can reduce the dimensionality of input features, thereby reducing the complexity and computational burden of the model, which may face dimensional disaster problems when the feature dimensionality is high, resulting in overly complex, overfitting, or computationally inefficient models [68]; it can also select features that contribute more to the model performance, thereby improving the model's prediction accuracy, precision, and other performance indicators; it can also select features that contribute more to the model performance, thereby improving the model's prediction accuracy and precision [69].

GEE provides an official technique for analyzing the importance of features, the ".explain()" method in classifiers, to obtain information about the importance of each classified feature. This procedure provides a quick measure of the importance of a variable, with a score indicating the magnitude of the importance. That is, the greater the importance, the higher the score. The importance value is standardized, usually corresponding to the ranked importance.

### 3.5. Supervision Classification

#### 3.5.1. Classifier and Parameter Settings

Random forest is a machine learning method based on integrated learning for classification and regression problems. It consists of multiple decision trees to build a powerful classification model by randomly sampling training data and selecting features. The parameter settings of random forest need to be tuned according to the specific problem and data set, and different parameter combinations may have different effects on the performance and generalization ability of the model. Of these, the increase in the number of decision trees can increase the complexity and prediction accuracy of the model. However, this may increase the computation time, and the value is too small to be easily under-fitted, so parameter tuning is needed to select an appropriate value. In this study, the parameters are tuned using cross-validation and other methods to select the appropriate number of decision trees and other parameters are taken as default values [25].

#### 3.5.2. Bare Land and Permafrost Classification

Bare land classification uses a hierarchical extraction strategy combined with machine learning to perform the following actions: remove water bodies, snow, and ice; classify bare land vegetation and built-up areas; remove built-up areas; and then calculate the annual maximum vegetation cover and select a suitable vegetation cover threshold to extract the extent of bare land. For permafrost classification, existing multi-source permafrost data, expert knowledge, and training sample refinement rules are combined to generate stable permafrost training samples, and machine learning algorithms are used to classify the permafrost.

#### 3.5.3. Classification Post-Processing

All pixel-based classification methods produce tiny patches in the classification results, which degrade the quality and accuracy of the images and may interfere with subsequent applications and analysis; they must therefore be eliminated to reduce the noise in the classification results. Small patch processing can further optimize classification results by enhancing their quality, interpretability, and spatial consistency and making them more continuous and rational [70]. In this study, the image of the ground object is first smoothed, then a mask of the ground object that is smaller than 50 pixels is extracted, and a sliding window replaces the smooth image acquired through this mode.

### 3.6. Mapping Accuracy Evaluation

In this study, the confusion matrix method was used to evaluate the accuracy of bare permafrost extraction using the overall accuracy and Kappa coefficient as a measure [71].

$$OA = \frac{TP + TN}{TP + TN + FP + FN}, \tag{2}$$

$$K = \frac{OA - P_e}{1 - P_e}, \tag{3}$$

$$Pe = \frac{(TP + FN) \times (TP + FP) + (FN + TN) \times (TN + FP)}{N^2}, \tag{4}$$

Here, OA is the overall accuracy; K is the kappa coefficient; true positive (TP): predicts positive class as positive class number; true negative (TN): predicts negative class as negative class number; false positive (FP): predicts negative class as positive class number false alarm; and false negative (FN): predicts positive class as positive class number as the number of negative classes.

## 4. Results and Analysis

### 4.1. Bare Permafrost Extraction on the Tibetan Plateau

4.1.1. Feature Selection for Bare Permafrost Extraction on the Tibetan Plateau

The importance score of the feature vector was calculated in this study using the feature importance analysis method officially supplied by GEE (Figure 4), based on the wave spectrum features, texture features, and terrain features we constructed. The features were ranked according to the importance score, and the features were added in order of importance from greatest to smallest in order to acquire the feature selection results.

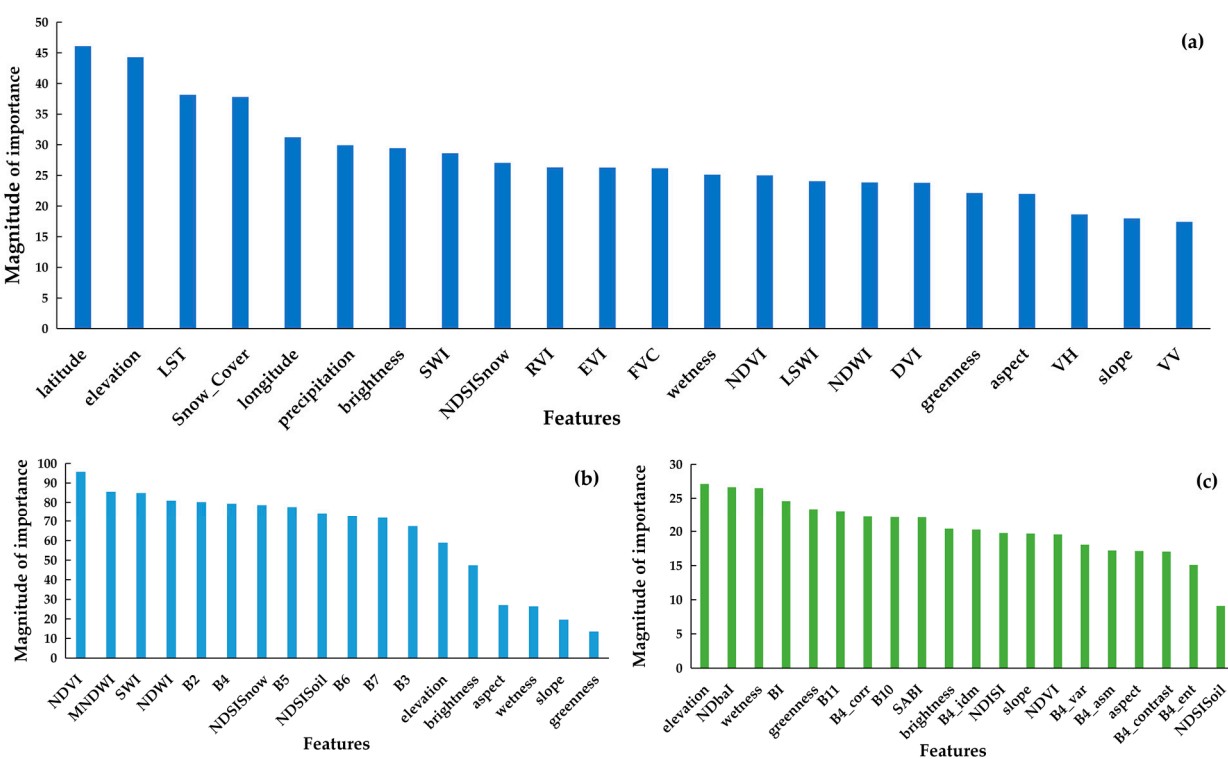

**Figure 4.** (**a**–**c**) are the importance of classification features for permafrost; the importance of the first classification feature for bare land; and the importance of the second classification feature for bare land, respectively.

When extracting permafrost, the mapping accuracy reaches the maximum when the number of features is 22. After that, the mapping accuracy stabilizes and decreases as the number of features increases. Due to the strong correlation of various types of vegetation features in permafrost extraction, the complexity of the model increases, so when extracting, we removed the vegetation features that had a minor impact on the mapping accuracy (EVI, FVC, RVI) and used the first 19 features as the features of permafrost extraction. The overall accuracy of the model was 0.862. When classifying bare land, the classification accuracy

was highest when the number of input features was 16 and 18; thus, the overall accuracy of the model was 0.972 and 0.93 when using the first 16 and 18 features, respectively. The results of feature selection are shown in Table 4.

**Table 4.** Feature selection results.

|  | Spectrum Characteristics | Texture Characteristics | Geographic Characteristics | Climate Characteristics |
|---|---|---|---|---|
| Permafrost characteristics | Brightness, SWI, NDSI$_{Snow}$, wetness, NDVI, LSWI, NDWI, DVI, greenness, VH, VV | None | Elevation, slope, aspect, longitude, latitude | LST, precipitation, Snow_Cover |
| Bare land characteristics | B2~B7, NDVI, MNDWI, SWI, NDWI, NDSI$_{Snow}$, NDSI$_{Soil}$, brightness, wetness | None | Elevation | None |
|  | B10, B11, NDbaI, wetness, brightness, greenness, BI, SABI, NDISI, NDVI | Second-order moments, contrast, correlation, variance, inverse moments | Elevation, slope, aspect | None |

### 4.1.2. Optimization of Random Forest Parameters for the Tibetan Plateau

In order to determine the optimal number of decision trees for random forest classification, the number of trees was initially set to 10, and the mapping accuracy was recorded. When the number of trees was increased by 5, the mapping accuracy was evaluated. The mapping accuracy improved dramatically as the number of trees increased (Figure 5). When the number of permafrost classification trees reached 80 and the number of bare land classification trees reached 60 and 70, the mapping accuracy reached its maximum and stabilized as the number of trees increased.

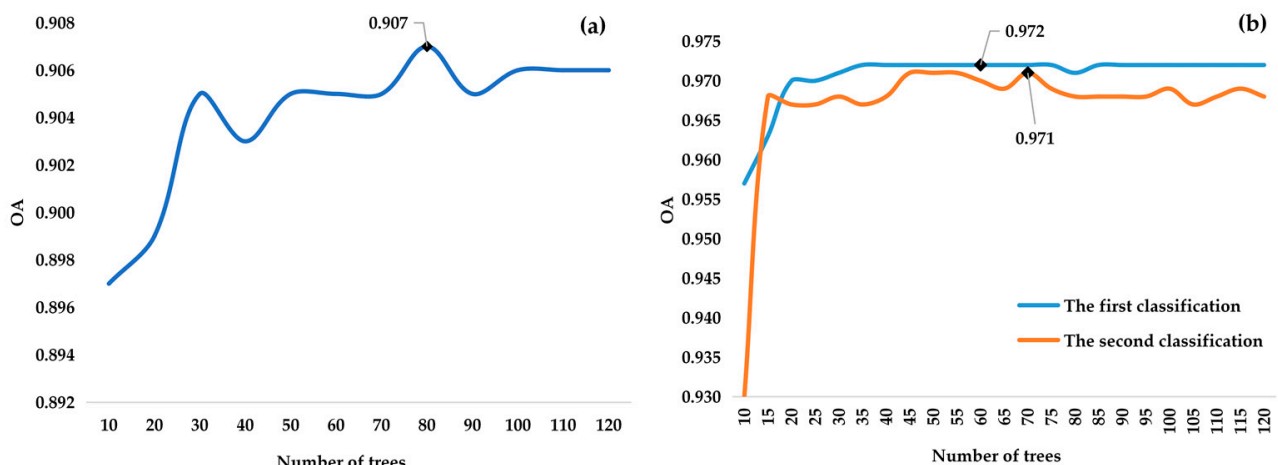

**Figure 5.** (**a**) shows the number of random forest trees for permafrost classification versus mapping accuracy, and (**b**) shows the number of random forest trees for bare land classification versus mapping accuracy (OA refers to the overall accuracy of the mapping).

### 4.2. Distribution Pattern of Permafrost on the Tibetan Plateau

#### 4.2.1. Reliability Analysis of Permafrost Training Samples

In this study, the overall accuracy of the training samples was 92.06 percent. The previous study by Zhang et al. [72,73] discovered that the overall accuracy tends to stabilize when the proportion of incorrect training samples is kept within a certain threshold and then swiftly decreases when the threshold is exceeded. We randomly altered the labels of a percentage of training samples in 1% increments to progressively increase the number of "contaminated" samples. Then, we used these samples to construct the RF classification model. Figure 6 depicts the quantitative relationship between mapping precision and error samples after 60 repetitions of this process. The overall and produced accuracy levels of permafrost classification are insensitive to error samples when the error sample percentage

is kept below 25%. The accuracy gradually decreases as the number of error training samples rises above this threshold. Therefore, this study's permafrost training samples are precise enough to facilitate permafrost mapping.

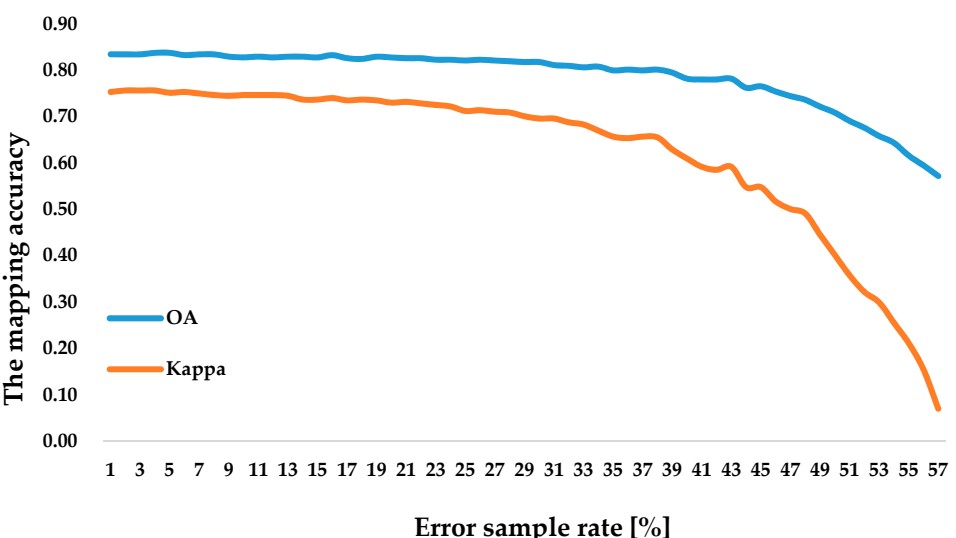

**Figure 6.** Relationship between error samples and mapping accuracy.

4.2.2. Comparative Analysis of Different Permafrost Extraction Methods

In conducting bare permafrost extraction, much information is not available due to the small amount of permafrost data on the Tibetan Plateau. The classification accuracy of existing permafrost products is low, and there is a general overestimation at the permafrost boundary. In order to solve the above problems, the optimal extent of permafrost was selected to reduce its influence on bare permafrost extraction. In this study, two commonly used machine learning methods, namely the random forest and support vector machine methods (Median_RF, Mean_RF, Median_SVM, Mean_SVM), in conjunction with median synthetic imaging, mean synthetic imaging, and five methods of mean ground temperature model (MAGT), were compared. To establish a complete dataset of permafrost distribution on the Tibetan Plateau from 2015 to 2017, the mean ground temperature model uses 360 permafrost sites of 10 m locations during the same years [74]. The ground temperature is used as the annual mean ground temperature, and the same characteristic variables are selected for the stepwise regression of the annual mean ground temperature, with the annual mean ground temperature of 0 degrees as the threshold [13]. The permafrost was extracted and validated with the remaining sample points, and the mean ground temperature model regression results were calculated as follows:

$$MAGT = LST \times 0.114 - elevation \times 0.004 - latitude \times 0.65 - DVI \times 17.185 + 38.73 \left( R^2 = 0.44 \right) \quad (5)$$

The choice of kernel function for the support vector machine impacts the model's classification performance and computational efficiency. In this investigation, the SVM kernel function was set to the Gaussian radial basis function (RBF), and the other parameters were set to their default values [75,76]. The permafrost extraction's overall accuracy and Kappa coefficients were calculated independently, as shown in

As shown in Figure 7, Median_RF has an overall accuracy of 90.79% when extracting permafrost on the Tibetan Plateau, with a Kappa coefficient of 0.806, which is the best performance in permafrost information extraction compared with other methods, and it can realize the high-precision extraction of permafrost on the Tibetan Plateau. Different permafrost extraction methods and image synthesis methods will affect the permafrost extraction results; in permafrost extraction, random forest classification is better than the support vector machine and average ground temperature models, median image synthesis

is better than mean image synthesis, and the effect of extraction methods on permafrost extraction results is greater than the effect of image synthesis. Figure 7.

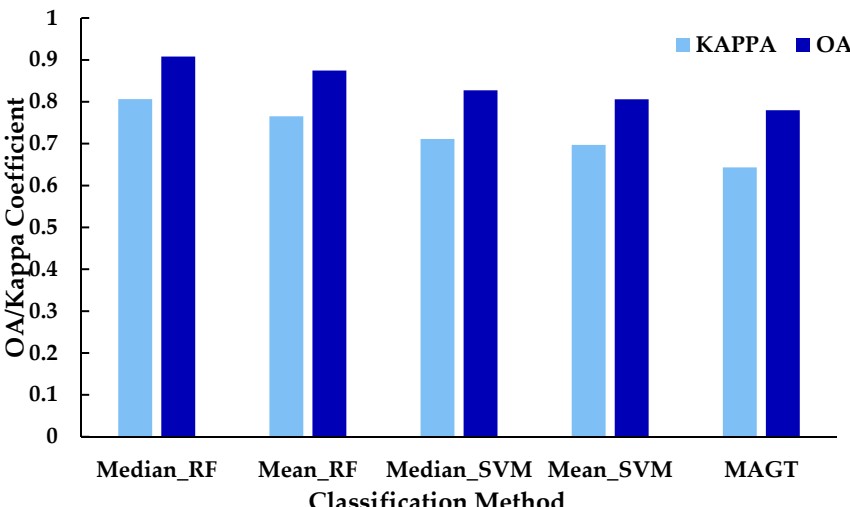

**Figure 7.** Comparison of median composite random forest method (Median_RF), mean composite random forest method (Mean_RF), median composite support vector machine method (Median_SVM), mean composite support vector machine method (Mean_SVM), and mean ground temperature model (MAGT) extraction results (OA is the overall accuracy of the mapping).

This study compared different methods of extracting permafrost using the same image synthesis (Figure 8). The results showed that Median_RF and Median_SVM were able to accurately exclude water bodies from permafrost extraction. However, in areas with fragmented permafrost distribution, Median_SVM and MAGT methods may overestimate the extent of permafrost. Median_RF performed best in snow- and ice-covered areas. The dataset used in the study was the 2005 permafrost distribution map, and Median_RF had a lower permafrost area in fractionally distributed permafrost extraction, possibly due to permafrost degradation.

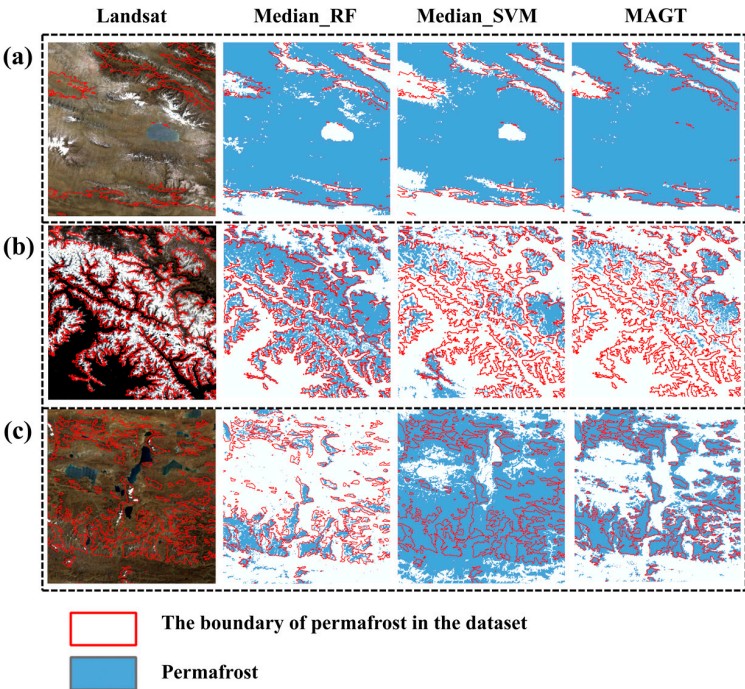

**Figure 8.** (**a–c**) are the Landsat images and the fine-grained results of Median_RF, Median_SVM, and mean geothermal model extraction for different locations on the Tibetan Plateau, respectively.

4.2.3. Permafrost Distribution

A map of the permafrost distribution is shown in Figure 9. When performing supervised classification sample collection, approximately 70% were selected as training samples and approximately 30% as validation samples, with an overall accuracy of 90.79% and a Kappa coefficient of 0.806. Generally, the perennial permafrost on the Tibetan Plateau is distributed over 2000 m above sea level, mainly in the range of 4000 to 5500 m, with the broadest distribution around 5000 m. At the same time, the elevation change in permafrost distribution is also related to the slope direction. The lower the altitude, the more pronounced the role of slope direction. The area of perennial permafrost on shaded slopes is larger than that on sunny slopes because the former receives less solar radiation than the latter. Above 5100 m above sea level, permafrost distribution is controlled by altitude, and the role of slope orientation is no longer prominent.

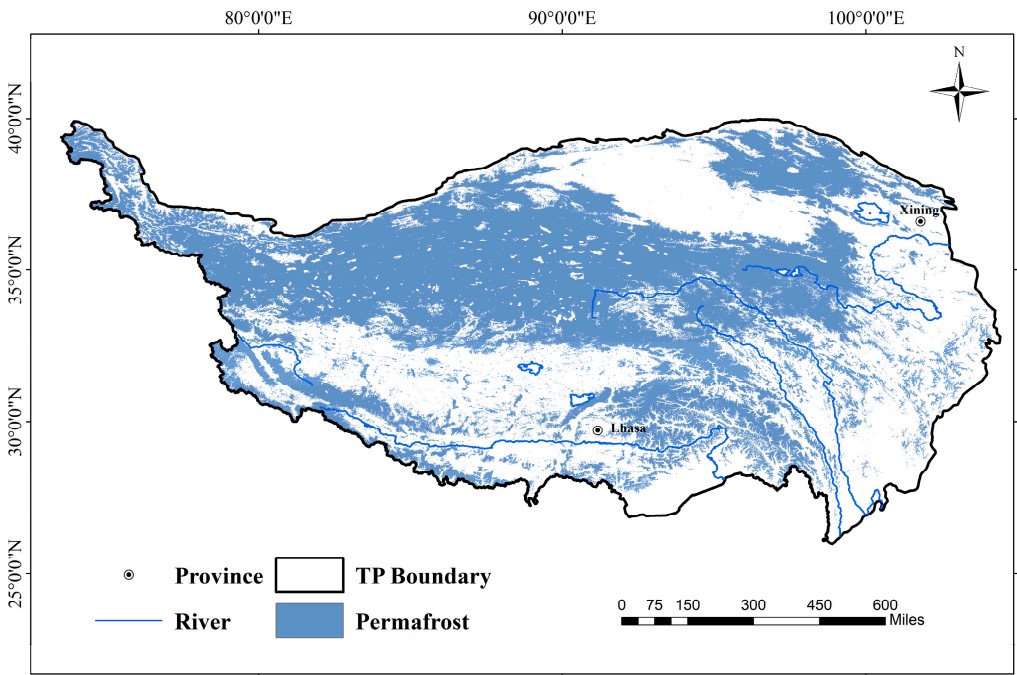

**Figure 9.** Map of permafrost classification results.

*4.3. Distribution Pattern of Bare Land on the Tibetan Plateau*

4.3.1. Effect of Fractional Vegetation Cover on Bare Land Extraction on the Tibetan Plateau

Typically, the fractional vegetation cover of bare land is regarded as an empirical threshold value. This threshold may differ slightly and be subjective based on the disparities in remote sensing data and vegetation types between regions [77]. Since 10–15% is generally regarded as the fractional vegetation cover of bare land [78], this threshold value of 5–15% fractional vegetation cover was chosen for the barren land in this study and compared to the land use dataset and land cover dataset of the Tibetan Plateau in the same year in order to determine the optimal threshold value of fractional vegetation cover for the bare land. The land use dataset and the land cover dataset of the Tibetan Plateau in the same year revealed that the bare land area was 459,200 km$^2$ and 446,600 km$^2$, respectively, primarily distributed in the northern, northwestern, and southern parts of the Tibetan Plateau, and the comparison with the existing dataset revealed that (as seen in Figure 10) when the fractional vegetation cover threshold was 12%, the area was similar and the spatial similarity was the highest (spatial similarity is defined in this study as the ratio of bare area to total bare land area in the same area) [79].

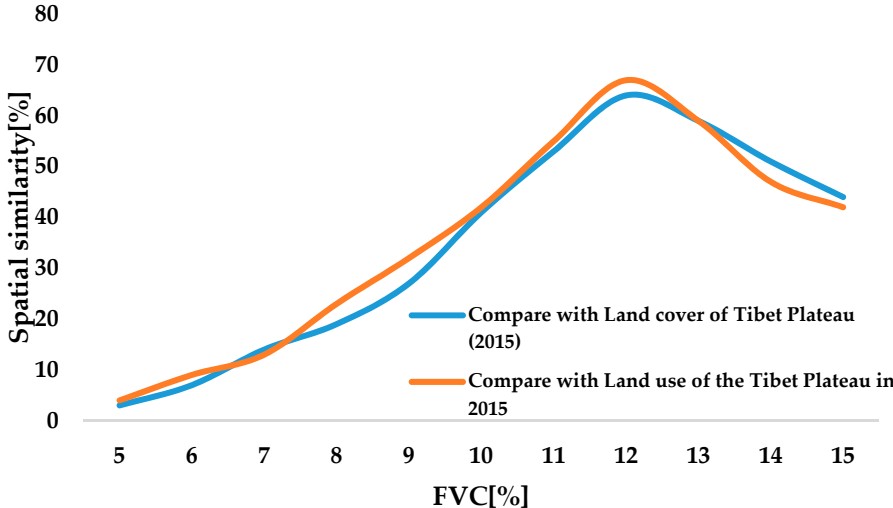

**Figure 10.** Comparison of the spatial similarity between the extraction results of bare land with different fractional vegetation cover and available information.

### 4.3.2. Bare Land Distribution

The process of extracting bare land was performed in layers. First, we classified water bodies, snow, ice, and built-up vegetation areas. Then, we classified bare land vegetation and built-up areas and excluded built-up areas. Finally, we calculated the annual maximum fractional vegetation cover and selected 12% [8] as the threshold value to determine the extent of bare land (Figure 11). During the classification, 70% were labeled as training samples, while 30% were classified as validation samples. The accuracy of the first group was 97.47% with a Kappa coefficient of 0.954, and the accuracy of the second group was 96.99% with a Kappa coefficient of 0.911.

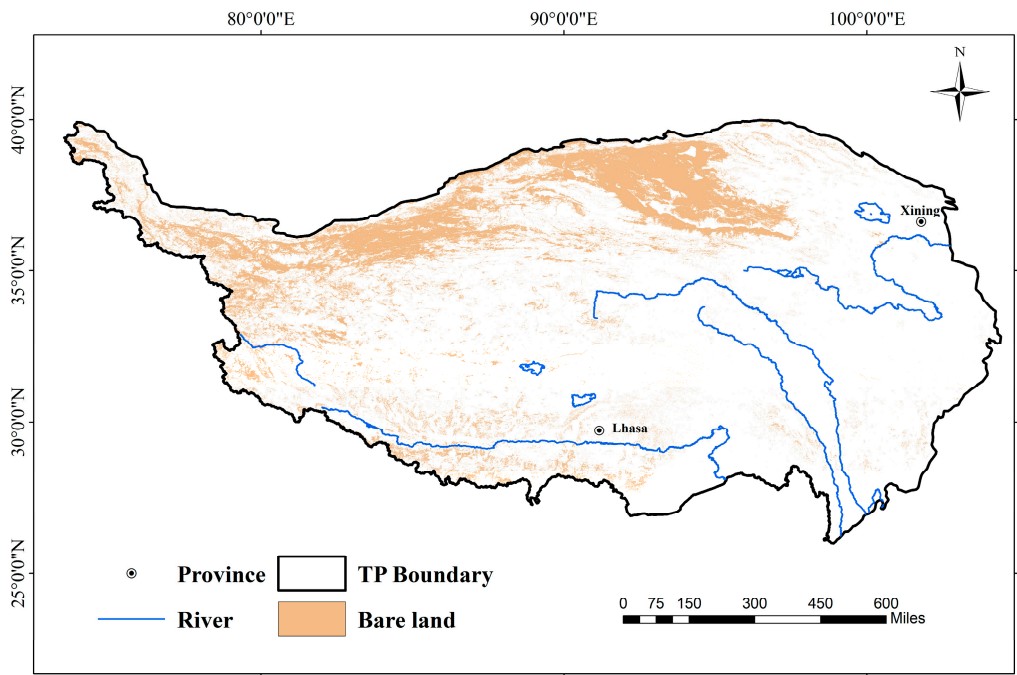

**Figure 11.** Map of bare land classification results.

### 4.4. Distribution Pattern of Bare Permafrost on the Tibetan Plateau

By obtaining the range of bare land and permafrost, the bare permafrost area can be identified via overlaying (Figure 12), and, according to the extraction results of this study,

the bare area of the Tibetan Plateau from 2015 to 2017 is about 462,000 km$^2$. The permafrost area of the Tibetan Plateau is about 1.16 million km$^2$, and the bare permafrost area is about 176,700 km$^2$. The majority of bare permafrost is found between 35° and 37.5°N latitude and below 90°E longitude, with most inclinations between 0° and 20°.

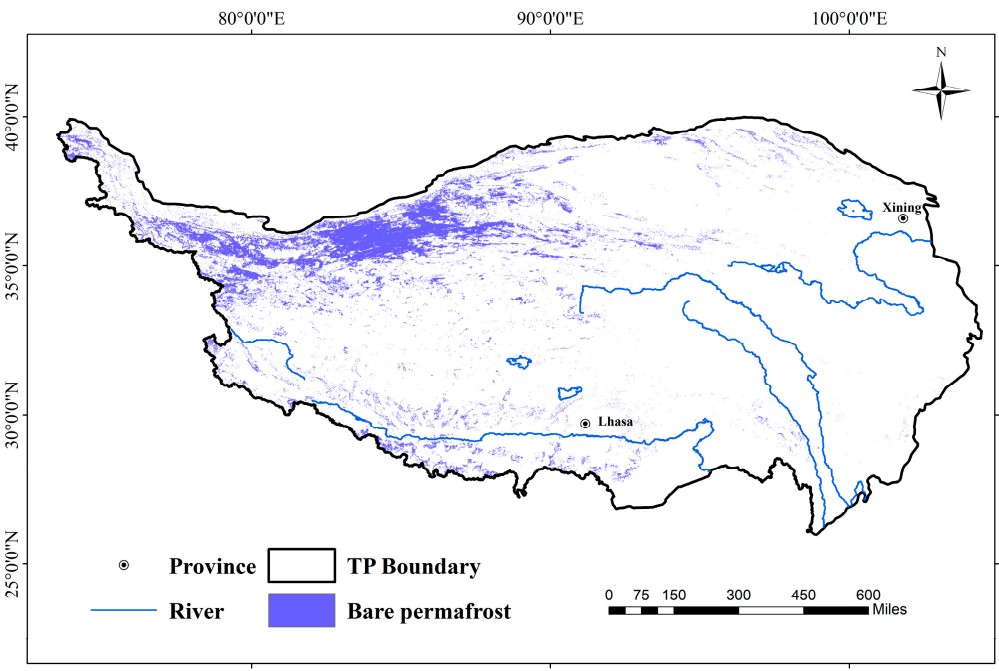

**Figure 12.** Map of bare permafrost classification results.

## 5. Discussion

### 5.1. Cross-Comparison with other Tibetan Plateau Permafrost Maps

Except in a few built-up areas, there is no direct evidence of extensive permafrost degradation on the Tibetan Plateau. Therefore, we disregarded the potential influence of differences resulting from study period and selected multiple datasets of permafrost distribution on the Tibetan Plateau to compare the areas that they classify as permafrost (Table 5, Figure 13).

**Table 5.** Multi-year permafrost areas on the Tibetan Plateau from different sources.

| Multi-Year Permafrost Maps | Area (10$^4$ km$^2$) | Source |
|---|---|---|
| Map of the current distribution of permafrost on the Tibetan Plateau | 111.3 | Niu, Fu-Jun, Yin, and Guo-An, 2018 [80] |
| Newly mapped permafrost distribution on the Tibetan Plateau | 106 | Zhao, Lin, et al., 2017 [37] |
| China's ice and snow permafrost map at a scale of 1:4 million | 154.25 | Schiavone and Middleson, 1988 [81] |
| Permafrost map of the Tibetan Plateau at a scale of 1:3 million | 122 | Cheng, G., and Li, Shude, 2011 [82] |
| Multi-year permafrost stability distribution map of the Tibetan Plateau | 115.02 | Ran Youhua et al., 2021 [39] |
| Probability map of multi-year permafrost at a 1 km resolution of the Tibetan Plateau | 117 | Cao, B., et al., 2019 [38] |
| In this study, the permafrost range of the Tibetan Plateau was examined | 116 (104.62~125.39) | This study |

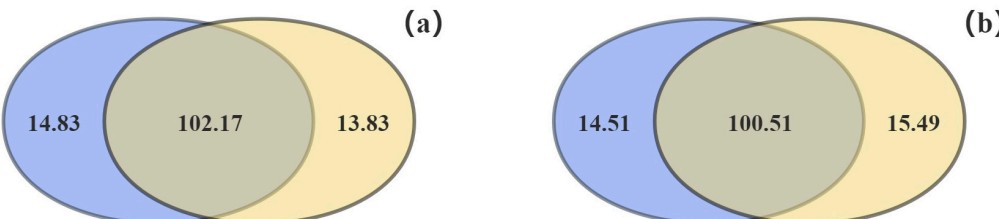

**Figure 13.** Comparison of extraction results of different methods. (**a**) Comparison between the results obtained in this study and the 1 km resolution permafrost probability map of the Tibetan Plateau. (**b**) Comparison between the results obtained in this study and the distribution map of permafrost stability on the Tibetan Plateau. Blue sections indicate permafrost in the dataset, yellow sections indicate extracted permafrost, and intersecting sections indicate correctly extracted permafrost. The number above each section indicates the permafrost area in a section of 10,000 square kilometers.

The differences found in China's 1:4 million snow and ice permafrost map are likely due to differences in the scale of mapping and the classification system used. To a lesser extent, the availability of data and mapping techniques, such as the extent of permafrost in the early years, was determined empirically based on the authors' understanding of limited information on topography and temperature. This may also result in substantial estimates of permafrost areas [83]. The permafrost map of the Tibetan Plateau at a scale of 1:3 million was created based on years of research on permafrost investigations and previous research papers, studies, and permafrost maps. In addition, its permafrost expanse is more precise, suggesting that more accurate results can be generated from existing information on the extent of permafrost and that the results should be more objective with the widespread use of GIS technology in post-2000 mapping [84,85]. This research incorporates more representative ground-based observations affecting permafrost with high-quality satellite data, previous research findings, and machine learning algorithms, so the multi-year permafrost given in this paper should be more credible.

### 5.2. Analysis of the Spatial Distribution of Bare Permafrost on the Tibetan Plateau

Three-way zonation, encompassing latitude, longitude, and altitude, also has a significant influence on the distribution and change in bare permafrost, and the spatial distribution characteristics of bare permafrost were analyzed by identifying the area of bare permafrost at different latitudes, longitudes, and altitudes (Figure 14).

It was found that the distribution of bare permafrost on the Tibetan Plateau had a strong three-way zonality, and the distribution at elevation is similar to a normal distribution, with the bare permafrost area increasing with the increase in elevation below 4500 m and decreasing after 5500 m. Moreover, most of the bare permafrost is concentrated in the area between 35 degrees and 37.5 degrees latitude, which accounts for about 61% of the overall area, as well as in the low longitude area of the Tibetan Plateau, where the area below 90 degrees longitude accounts for about 88% of the bare permafrost area.

Slope and aspect data were calculated from the DEM data, and they were superimposed with the bare permafrost extraction result data in order to analyze the spatial distribution characteristics of the bare permafrost (Figure 15).

The slope of bare permafrost in the study area is mainly concentrated in the range of 0°~20°, and the area of bare permafrost in this range accounts for more than 96%, which is due to the slope being too large for factors such as the ground brightness and the ground humidity to have an effect, thus influencing the formation of bare permafrost. Regarding aspect, the distribution of bare permafrost is consistent between shaded slopes and sunny slopes, with the highest distributions on slopes facing north and south, which are 14.4% and 16.5%, respectively. Aspect affects many factors, such as water evaporation, vegetation cover, and the erosion of slopes, which results in the distribution of bare permafrost at the same altitude and in different aspects.

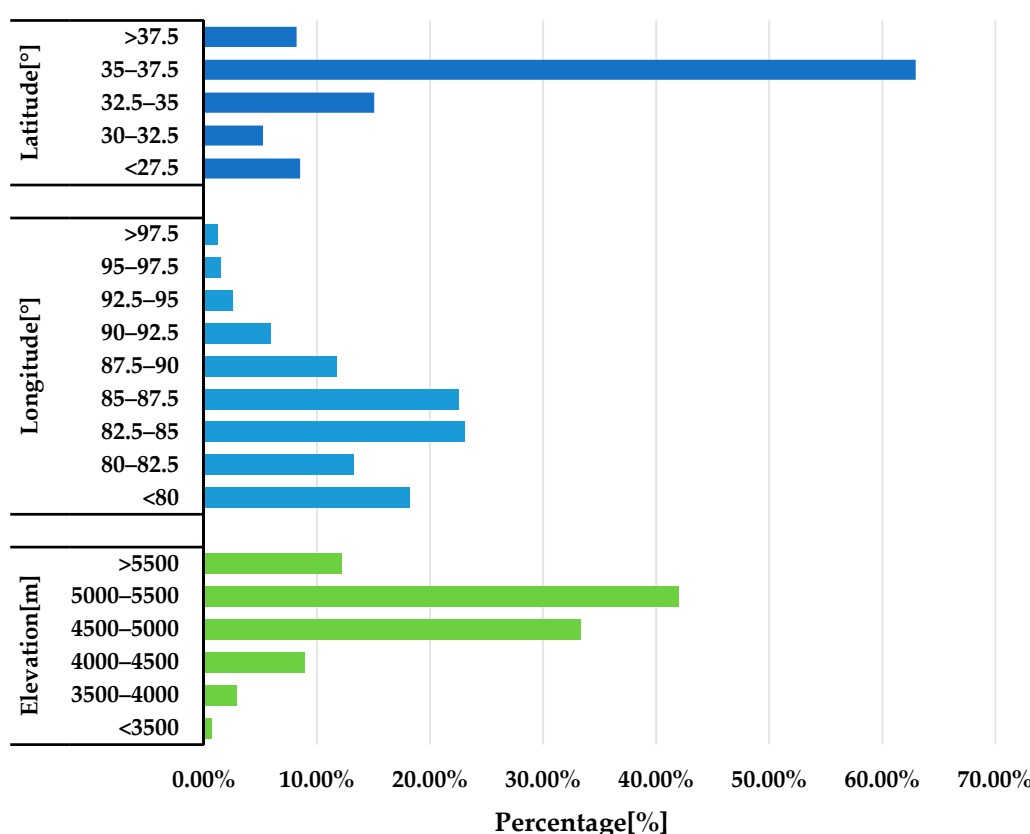

**Figure 14.** Percentage of bare permafrost at different elevation, longitude, and latitude.

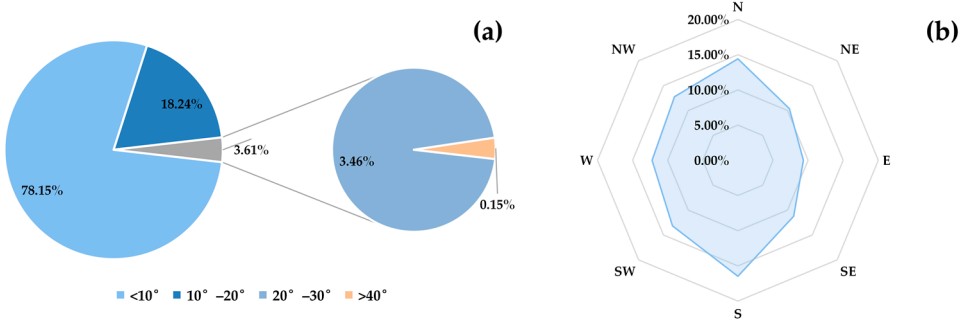

**Figure 15.** Percentage of bare permafrost on different slopes (**a**) and at different aspects (**b**).

### 5.3. Influence of Climatic Factors on the Distribution of Bare Permafrost

As bare permafrost refers to permafrost with little or no vegetation on the surface, the impacts on both bare land and permafrost can directly affect the distribution of bare land. On the one hand, climatic factors such as precipitation and temperature can lead to changes in the distribution of bare land; dry climates usually have more bare land as precipitation is not sufficient to sustain vegetation growth and climates with little rainfall can also lead to drought and the evaporation of the soil, thus exposing the land. Climatic conditions of extremely high or low temperatures may limit vegetation growth and result in exposed land [86]. High temperatures can trigger evapotranspiration, which dries out the soil, while low temperatures can lead to chilling, which damages vegetation. Strong winds can blow away vegetation cover and expose the land. In addition, winds can influence soil erosion processes, further exposing bare land [87].

On the other hand, various climatic factors can affect changes in permafrost. Surface temperature affects the distribution of permafrost by acting on the process of water–heat transfer between the atmospheric surface and thus affects the distribution of perennial

permafrost; there are significant differences in the effects of snow cover on shallow ground temperatures in different regions of the plateau [43]. The high albedo of the snow can reduce the ground temperature and weaken the intensity of soil-freezing in the cold season. The snow's larger heat capacity reduces the intensity of heat conduction. Snow cover can also delay the impact of external climate change on permafrost to a certain extent. Precipitation can change the surface energy balance and soil thermal parameters to achieve changes in the freezing and thawing process of the soil and the process of water–heat transport, thus affecting the hydrothermal stability of permafrost [19]. It can also increase the temperature of the active layer and permafrost by increasing the soil's water content, elevating the thermal conductivity of the soil layer, and increasing the heat transfer from the surface and the latent heat of thawing of the soil, which will further affect the distribution of permafrost [88]. In summary, climate has an enormous impact on the distribution of bare permafrost, and low temperature, precipitation, seasonal temperature changes, and climate change all have an essential impact on the existence and stability of bare permafrost. Therefore, studying the relationship between bare permafrost distribution and climate is vital for understanding the evolution of bare permafrost, managing groundwater resources, and addressing the challenges posed by climate change.

## 6. Conclusions

In this study, we proposed a more reliable permafrost sample point selection method using meteorological, spectral, and other multi-source data combined with limited permafrost survey data. By comparing the optimal image synthesis method and the machine learning method and obtaining a multi-year permafrost map of the Tibetan Plateau, the validation results show that the overall accuracy of this paper's method in permafrost extraction on the Tibetan Plateau is 90.79%. The Kappa coefficient is 0.806, indicating that the method has good mapping results and a high level of accuracy. Meanwhile, the layered extraction strategy combined with machine learning and the existing Tibetan Plateau land use data was used to determine the vegetation coverage threshold for bare land extraction, and thus, the extent of bare land on the Tibetan Plateau was extracted, which had a high spatial similarity measure when compared with the existing land use data on the Tibetan Plateau, in order to construct a bare permafrost extraction method based on the fusion of multi-source machine learning information on the Tibetan Plateau. The bare permafrost extraction method proposed in this study can be used for bare permafrost mapping in other alpine regions. The classification method has the advantage of being transferable to other regions and datasets at the same time. This dataset can be used as an accurate map of bare permafrost through the accurate extraction of bare permafrost expanse, which can provide technological support for obtaining information on the distribution of bare permafrost in the Tibetan Plateau over a long period and for revealing the impacts of climate change on the bare permafrost in the Tibetan Plateau.

**Author Contributions:** Conceptualization, X.L. (Xiaoyang Li) and Y.J.; methodology, G.Z.; validation, X.L. (Xiaopeng Li), G.Z. and Y.J.; formal analysis, G.Z. and L.Z.; data curation, Y.J.; writing—original draft preparation, X.L. (Xiaoyang Li); writing—review and editing, Y.J.; funding acquisition, G.Z.; supervision, X.H. and Z.T. All authors have read and agreed to the published version of the manuscript.

**Funding:** This work was supported by the Second Tibetan Plateau Comprehensive Research Project (2019QZKK0106), the National Natural Science Foundation of China (42130514), the Fundamental Research Funds of the Chinese Academy of Meteorological Sciences (2023Z023, 2022Y015), and the Meteorological Satellite Engineering Project (FY-APP-2022.0309).

**Data Availability Statement:** Data are contained within the article.

**Acknowledgments:** We thank the Google Earth Engine Science team for the freely available cloud-computing platform; ESA for Sentinel-1 imagery; USGS for Landsat imagery and SRTM DEM; UCSB for CHIRPS Daily; and NASA for MODIS. We thank the National Tibetan Plateau Data Center for providing the permafrost and bare land datasets.

**Conflicts of Interest:** The authors declare no conflict of interest.

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
