# Peer review of "A New Method for Bare Permafrost Extraction on the Tibetan Plateau by Integrating Machine Learning and Multi-Source Information"

_remotesensing, doi:10.3390/rs15225328_

Round 1

Reviewer 1 Report (Previous Reviewer 1)

Comments and Suggestions for Authors

1.       Improvements to the previous manuscript have met expectations, except for some figure quality, which needs to be improved.

1   2.     Please check the quality of the figures in your paper again, especially Figure 1, and you must replace it with a better layout and image quality.

Author Response

Reviewer 2 Report (Previous Reviewer 2)

Comments and Suggestions for Authors

I reviewed the maunscript but my previous concerns  have not been solved, although I found the authors did their best to revise the manuscript.  The major comments are listed as follows:

1. The motivation of this study is not clear. I suggest the authors should refer to more recent references.

2. The blurred images should be improved.

3. Authors declared that they aimed to produce a reliable permafrost map with 30m resolution. However, the input datasets with coarse resolution (500m-5km) were just roughly resampled to  30m with unkonwn methods. This process is suspect in my practice. Based on this view, I think the results are unbelievable .

Comments on the Quality of English Language

English must be edited and improved. 

Author Response

Reviewer 3 Report (Previous Reviewer 4)

Comments and Suggestions for Authors

The revision has well covered all my comments, good job. I think it is quite ready for publication in current version.

Author Response

Thank you so much for reviewing our manuscript.

Round 2

Reviewer 2 Report (Previous Reviewer 2)

Comments and Suggestions for Authors

This manuscript can be accepted in its present condition.

Author Response

Thank you so much for reviewing our manuscript.

This manuscript is a resubmission of an earlier submission. The following is a list of the peer review reports and author responses from that submission.

Round 1

Reviewer 1 Report

Comments and Suggestions for Authors

1.    The explanation regarding the novelty is not clear. Please explain in more detail.

2.    You mention a lot of data as a source of information, including sentinel 1A data processing, but it is not found in the research flowchart in Figure 2. How can you explain?

3.    How do you analyze synthetic aperture radar data to identify bare permafrost? Please explain in your paper.

4.       From Table 2, it appears that you are mainly using optical data. How do you deal with cloud cover?

5.       In Table 3, you mention the related spectral characteristics of Sentinel 1 VV and VH. What do you mean by spectral characteristics?

6.       In the discussion and conclusion section, you should highlight the novelty aspect of the method, not the discovery of bare permafrost.

Reviewer 2 Report

Comments and Suggestions for Authors

In this study, Li et al., aims to apply a new method for bare permafrost extraction for the Tibetan Plateau (TP). They use machine learning model of permafrost occurrence. The study is said to represent “a new method” modeling that can reveal “bare permafrost” distribution. Despite the potential importance of the permafrost map for the TP, there are number of issues that the precludes me to recommend the acceptance of this manuscript in its current form. My main criticism concerns the added value of this work in relation to the existing literature. The authors claim the modeling to be “a new method”, when actually similar modeling, but with substantially larger hemispherical extents, have been published recently (e.g. Obu et al., 2019; Aalto et al., 2018; Ran et al., 2021). Therefore, it’s far for being clear how much the present work actually improves the representation of permafrost over this climatically sensitive region. The methods are widely used and not so novel. The study would benefit from a proper uncertainty analysis, which could tell us something about the impacts of different data and modeling choices made along the way.  Although I am not a native English speaker, it’s evident that there are a lot of problems with the language and general organization of the text, and for that reason the manuscript is often quite difficult to follow. I also feel that there is a certain amount of hastiness with a clear lack of finishing touch. Meanwhile, I am confused with the concept of “bare permafrost”. Sorry for the above straightforward comments, but authors should really update their literature review to reflect advances in this field.

References:

Aalto, J., Karjalainen, O., Hjort, J., Luoto, M., 2018. Statistical forecasting of current and future circum-Arctic ground temperatures and active layer thickness. Geophys. Res. Lett. 45 (10), 4889–4898.

Obu, J., Westermann, S., Bartsch, A., Berdnikov, N., Christiansen, H.H., Dashtseren, A., Delaloye, R., Elberling, B., Etzelmüller, B., Kholodov, A., Khomutov, A., Ka¨ab, ¨ A., Leibman, M.O., Lewkowicz, A.G., Panda, S.K., Romanovsky, V., Way, R.G., Westergaard-Nielsen, A., Wu, T., Yamkhin, J., Zou, D., 2019. Northern Hemisphere permafrost map based on TTOP modelling for 2000-2016 at 1 km2 scale. Earth Sci. Rev. 193, 299–316.

Ran, Y., Li, X., Cheng, G., Che, J., Aalto, J., Karjalainen, O., Hjort, J., Luoto, M., Jin, H., Obu, J., Hori, M., Yu, Q., Chang, X., 2022b. New high-resolution estimates of the permafrost thermal state and hydrothermal conditions over the Northern Hemisphere. Earth Syst. Sci. Data 14 (2), 865–884.

Comments on the Quality of English Language

Extensive editing of English language required

Reviewer 3 Report

Comments and Suggestions for Authors

The authors propose a method for identifying frozen soil through machine learning based on multiple remote sensing indicators in this paper. The article can be improved in three areas: (1) The formula in the remote sensing indicator list does not provide the most original literature sources, and there is a lack of explanation of the interrelationships between indicators, such as the applicability or advantages of the two water body indicators NDWI/MNDWI. This flaw limits further discussion of the results and the direction of method optimization in the future. (2) The results analysis section of the article focuses on the comparison of some model attributes provided by machine learning, as well as the consistency of visual interpretation results of some special region images. Lack of more systematic comparison with other data, such as differences in soil movement distribution and interannual variation characteristics under different altitudes and vegetation types, in addition to area. Therefore, the conclusion of the article is weak and lacks in-depth discussion. (3) The article lacks design in the diagrams, both in the flow diagram of the method section and the diagram of the results section, indicating that there is no logical hierarchy analysis and comprehensive comparison of the diagrams, which requires overall optimization and improvement.

Comments on the Quality of English Language

I can understand the author's English expression, but it cannot be called fluent, accurate, or appropriate English expression.

Reviewer 4 Report

Comments and Suggestions for Authors

The study covers the extraction of bare permafrost in Tibetan Plateau. Machine learning such as random forest was introduced, multi-source indicators were also provided in this manuscript. Quite novel and important for the distribution of exposed permafrost. However, many revisions are still required before publication, my specific comments are as follows:

1. Rewrite the sentences from line 14 to line 16 for better understanding.

2. Line 30, two layers, what layers?

3. You pointed out the results in the study can facilitate the elucidation of the impact of climate change on exposed permafrost, it should be discussed in detail.

4. Tibetan Plateau in title, Qinghai-Tibet Plateau in the running text? you must unify these descriptions throughout the whole manuscript.

5. Line 42 and 43, too many furthermore, replace them with meanwhile, in addition et al.

6. Rewrite Line 45 to 50 for better understanding.

7. Line 54, you have defied bare permafrost in your study, by what criteria, references?

8. Line 92, delete redundant space.

9. Line 109, passive tense is suggested.

10. Line 118, change 3,083,400 with 3.08*106

11. Line 138, the resampling process should be introduced in detail.

12. 172 , a machine learning algorithm, what algorithm? RF?

13. . is missing in 196.

14. Too many regions were missing between original and open operation in figure 3, why?

15. Line 239-242, GLCM texture features on what computation scale (window size), what steps?

16. How RF was used for classification in this study, you should explain it more.

17. The parameters for RF?

18. Line 295, introduce more about the cross-validation and other parameters, too sketchy.

19. Delete the redundant space in line 296.

20. Line 327, importance analysis method should be clearly introduced in M&M section.

21. Discussion part, the comparisons between different methods should be clarified in introduction part (hypothesis of the study).

22. Replace Figure 9, Figure 10, Figure 11, Figure 12 and corresponding text into results.

23. The effect of permafrost distribution on ecology and climate change should be discussed in discussion part for better importance.

24. Only one paragraph is suggested in the Conclusion section.